# Effects of Augmented-Reality-Based Exercise on Muscle Parameters, Physical Performance, and Exercise Self-Efficacy for Older Adults

**DOI:** 10.3390/ijerph17093260

**Published:** 2020-05-07

**Authors:** Sangwan Jeon, Jiyoun Kim

**Affiliations:** 1Exercise Rehabilitation Convergence Institute, Gachon University191 Hombakmoero, Yeonsu-gu, Incheon 406-799, Korea; jsw3972@kspo.or.kr; 2Department of Exercise Rehabilitation & Welfare, Gachon University 191 Hombakmoero, Yeonsu-gu, Incheon 406799, Korea

**Keywords:** exercise self-efficacy, augmented-reality-based, muscle reduction prevention program, sarcopenia, Korean older adults

## Abstract

This study was intended to determine the applicability of an augmented-reality-based muscle reduction prevention exercise program for elderly Korean women by observing changes in exercise self-efficacy and verifying the effectiveness of the program in the elderly after the application of the program. A total of 27 participants, who were elderly women aged 65+ and had not participated in any exercise programs until this study, were recruited for this study. They were divided into an experimental group (13 people) and a control group (14 people), and then the augmented-reality-based muscle reduction prevention exercise program was applied. This was a 30-min program, which included regular, aerobic, and flexibility exercises, and it was applied 5 times a week for 12 weeks. As a result of observing changes, it was found that the appendicular skeletal muscle mass (ASM) (F = 11.222, *p* < 0.002) and the skeletal muscle index (SMI) (kg/m^2^) (F = 10.874, *p* < 0.003) muscle parameters increased more in the experimental group compared to the control group, and there was a significant increase in gait speed (m/s) (F = 7.221, *p* < 0.005). For physical performance, as a result of conducting the Senior Fitness Test (SFT), a significant change was observed in the chair stand test (F = 5.110, *p* < 0.033), 2-min step test (2MST) (F = 6.621, *p* < 0.020), and the timed up-and-go test (TUG) (F = 5.110, *p* < 0.032) and a significant increase was also observed for exercise self-efficacy (F = 20.464, *p* < 0.001). Finally, the augmented-reality-based exercise program in this study was found to be effective in inducing physical activity in the elderly. Therefore, the augmented-reality-based muscle reduction prevention exercise program is considered to be effective in increasing the sustainability of exercise, thus preventing muscle reduction in the elderly.

## 1. Introduction

The population of the elderly aged 65+ accounted for 14.4% of the total population in Korea in 2018, and this is estimated to grow to 41% of the total population by 2060 [1], indicating that the elderly population growth rate in Korea is unprecedented relative to other places in the world. In this context, sarcopenia is emerging as an important issue in modern society. In 2016, the World Health Organization (WHO) published the 10th edition of the International Statistical Classification for Diseases and Health Issues (ICD-10) and assigned a disease code (M62.84) to sarcopenia, which is a major step in recognizing it as a disease [2]. Sarcopenia refers to a condition in which muscle mass and muscle strength decrease and muscle quality decreases because of gradual skeletal muscle reduction in conjunction with aging [3]. Muscle mass decreases by about 1% per year in people over the age of 40 [4], and skeletal muscle loss in conjunction with a reduction of muscle mass and muscular strength is known to be accompanied by a decrease in gait speed [5].

Elderly people with sarcopenia are up to 1.5 times more likely to have a stroke and cancer. It not only increases the prevalence of obesity and chronic disease but also leads to a vicious cycle—physical disorders, poor quality of life, and even death [3]. Therefore, it is imperative that efforts are made to prevent muscle reduction in the elderly. As sarcopenia is caused by a lack of physical activity with advancing age [6], continued exercise is critical [7]. Aerobic exercise is recommended as a method of managing physical fitness and preventing sarcopenia and aging in the elderly, and resistance exercise is known to be effective not only in maintaining and strengthening muscle functions but also in preventing falls and the impairment of daily living in the elderly [8,9,10].

However, as the elderly have trouble continuing exercise because of physical and environmental restraints, it is very important to motivate them to continue to exercise and stay focused with purpose [11]. Therefore, it is necessary to introduce exercise contents in various ways for them to continue to exercise.

Augmented reality (AR) is a technology that combines the real world and 3D virtual object digital content with additional information into a single image in real time. In other words, augmented reality is a technology that fuses physical space in reality with cyber space and virtual data or objects, thus allowing interaction in real time [12].

In particular, the development and application of information and communications technology (ICT) in modern society allows diseases to be managed and health to be promoted in an effective way [13], and the AR system is used as a training tool to encourage users to continue rehabilitation through fun and motivation [14]. AR technology, in particular, is widely used in the rehabilitation of those with diseases such as stroke, because three factors of rehabilitation (repetition, rapid feedback, and motivation) conveyed through augmented reality have been found to be very effective [15].

In studies of augmented reality, participants became interested and motivated to exercise, and access to and control of the exercise became easy. As a result of the exercise, the participants could improve muscle strength, range of joint motion, speed of exercise, and ability to control exercise [14]. The feedback function of virtual reality is very helpful in retraining the movement of users by displaying information about errors in their movement and tracking correct movement [16]. In addition, AR, a field of virtual reality (VR), is a hybrid VR system that synthesizes virtual objects in a real environment and makes them look like the objects existing in the original environment, and augmented reality technology that mixes the real environment with virtual objects can provide a better sense of reality by allowing users to see the real environment, thus promoting and inducing rehabilitation in a real exercise rehabilitation environment [17].

However, as most of the augmented-reality-based exercise rehabilitation programs are conducted for patients with stroke, hemiplegia, and other conditions, the development and verification of programs that apply to the elderly are insufficient. In consideration of the effectiveness of the strength, gait, and motivation in patients, augmented-reality-based exercise rehabilitation programs would be effective even for the elderly who are less interested in participating in exercise or have difficulty in taking part in activities. Therefore, this study, by providing an augmented-reality-based exercise program that prevents muscle loss, aims to prevent rapidly decreasing muscle loss in the elderly and confirm the effect of augmented-reality-based exercise on the exercise self-efficacy in the elderly, thus exploring the sustainability of augmented-reality-based exercise.

## 2. Materials and Methods

### 2.1. Study Design and Participants

This study was designed to determine the applicability of an augmented-reality-based muscle reduction prevention exercise program for elderly Korean women by observing changes in exercise self-efficacy and verifying the effectiveness of the program in the elderly after its application.

Elderly women aged 65 or older who had never participated in physical exercise programs until then were eligible for this study, and among them only those who were above the cut-off value of 5.67 kg/m^2^, which is the criterion of the SMI used by the European Working Group on Sarcopenia in Older People (EWGSOP), were included in this study [3]. Those who had health problems (chest pain, dizziness, or a doctor’s recommendation on restricted exercise) and those who were not able to understand the purpose of the study, due to physical disorders, mental illness, and/or disability in cognition and communication, were excluded from the study. Only those who understood the purpose of this study and signed the consent form finally participated in this study. The sample size was calculated using the G-Power 3.1 program. The α level, test power, and effect size were set to 0.05, 0.80, and 0.80, respectively, so the minimum number of participants in this study was 15. However, in this study, 30 participants were recruited, which allowed for the potential dropout of some participants. Participants in the study were 30 elderly women who had attended the elderly welfare centers located in Incheon City from Jan 7 to Jan 18, 2019 and had been randomly selected in the allocation ratio by the SPSS Win23.0 program. They were randomized into the two groups—the experimental group (15 participants) treated with the augmented-reality-based exercise and the control group (15 participants)—at the allocation ratio of 50.0%. The control group was noncontact other than the participants being tested pre- and post-experiment, and participants in this group were asked not to do any exercise during the experiment. During the experimental period, two participants in the experimental group and one participant in the control group gave up for health reasons. The experiment was finally conducted with a total of 27 participants; there were 13 participants in the experimental group and 14 participants in the control group (Figure 1). The demographic characteristics of the participants are expressed in Table 1.

### 2.2. Augmented-Reality-Based Muscle Reduction Prevention Exercise Program

In this study, UINCARE-HEALTH^TM^ was used, which is a product that provides a VR-based exercise rehabilitation program that can interact with participants (UINCARE-82B, UINCARE corp., Korea). The system is shown in Figure 2.

The sensor it uses is a universal serial bus plug-and-play device that translates the scene geometry into depth information. From the point at which it is located, the sensor has an effective angle of 70°, a distance range of 1.2–4.5 m, and a response time of 10 ms and generates images of the participant with a resolution of 640 × 480 at 30 fps. A computer operated by Window 8.1 or later with a 3.1 GHz Intel Core i5 and 8 GB RAM renders the images onto a 40-in. (or larger) monitor with a resolution of 1920 × 1080.

This device, which is used for exercise rehabilitation to restore the reduced functions of the upper and lower limbs, serves as a kinetic-test evaluation device used to increase the range of joint motion in the exercise sites and applies a load by measuring and evaluating the pre-set joint angle and momentum. The device is configured with a general purpose PC, a 3D motion analysis sensor, and UINCARE software.

When users exercise while watching and following content displayed on the display of a general purpose PC, UINCARE’s 3D motion analysis sensor recognizes the patient’s body and motion in real time and measures the joint angle of each part of the body, such as the shoulder, elbow, wrist, trunk, neck, hip, knee, and ankle, to analyze the user’s motion.

The 3D motion analysis sensor attaches the virtual marker to the anatomical position of the patient by identifying the joint of the user’s body based on the depth image obtained by the time-of-flight (TOF) method. When the user performs an action, a RGB camera and an infrared camera analyze the patient’s motion in real time, capture it in the form of a media file, digitalize the position of the marker, and then analyze the motion information, such as the motion angle and angular velocity, for each joint in 3D form. The collected information, such as the patient’s joint angle and joint position, and the results of the exercise are displayed on the general purpose PC and are stored there for later management.

The UIN-HEALTH system is operated by the exercise specialist who has control of the participant’s training modules (content) and the level of difficulty. The experimental group individually participated in the exercise program at the time appointed with the exercise specialist, and the exercise specialist, serving as an advisor, selected the appropriate exercise for the participant, the output safety during the exercise, and the exercise feedback results (Figure 3).

The details of the exercise program are shown in Table 2. The augmented-reality-based muscular dystrophy prevention exercise program combines resistance, aerobic, and flexibility exercises. The exercise program was a 30-min augmented-reality-based exercise program configured with upper and lower body exercises and aerobic exercise and was conducted 5 times a week for 12 weeks.

### 2.3. Muscle Mass and Muscle Function

The standing height of the participants was measured to the nearest 0.1 cm with a stadiometer (Seca, Seca Corporation, Columbia, MD, USA). Body mass and lean muscle mass were measured to the nearest 0.1 kg with a Bioelectrical Impedance Analysis(BIA) (Inbody 720, Biospace, Seoul, Korea). In addition, the body fat percentage was estimated by the BIA. The European Working Group on Sarcopenia in Older People (EWGSOP) classifies low muscle mass and quality using a skeletal muscle index (SMI, ASM/height^2^). The appendicular skeletal muscle mass (ASM) was measured by the BIA (Inbody 720) and expressed as the sum of the muscle mass of the four limbs.

Gait speed and hand grip strength were used for muscle function. Gait speed (m/s) was calculated from the time required for participants to walk a 6-m course at their usual pace, and the average of two trials was used. For muscle strength, the score of the grip strength in the Senior Fitness Test was used.

### 2.4. Assessment of Physical Performance: Senior Fitness Test (SFT)

Physical performance was assessed by a physical performance test of the elderly at the national fitness awards. The national physical fitness test parameters yielded significantly consistent results, with reliability ranging from 0.62 to 0.93 [18]. All parameters were measured by professionally certified health and fitness instructors. Six assessments were conducted, including a hand grip strength test, a chair stand for 30 s, a 2-min step test (2MST), a sit and reach test (cm), a timed up-and-go test (TUG), and a figure-of-eight walk test (F8W).

Hand grip strength (kg): Hand grip strength was measured using a hand dynamometer (GRIP-D 5101, Takei, Niigata, Japan) placed between the fingers and palm at the base of the thumb in one hand, with the arm extended to the side; the participant squeezed the dynamometer with maximum force. Participants were instructed to stand in an upright position, keep their arms straight, maintain a 15° angle between the arm and torso, and squeeze the dynamometer for 5 s at maximal effort. After both the left and right hands had been measured twice, the highest value was recorded to the nearest 0.1 kg.

Chair stand test (number in 30 s): The participant stood up from and sat down on a chair that had a height of 44 cm. The trial was initiated by sitting on the chair, the feet resting on the floor, and the hands crossed at the wrists and held on the chest. On a command given by the examiner, the participant performed as many of the standing-up cycles as they could within 30 s. If the 30 s had passed while the participant was in the standing position, the stand-up cycle was considered complete and incorporated into the score. The number of performed cycles constituted the test result.

Two-minute step test (2MST) (number of steps): The participant stood up straight next to a wall, with the target level for their steps corresponding to midway between their patella and their iliac crest. The test required the participants to march in place for 2 min, lifting the knees to the marked target on the wall. The number of times the right knee met the marked target was counted. An increased step count within the 2 min was reflective of a greater cardiopulmonary endurance.

Sit and reach test (cm): The participants’ general flexibility was assessed with a sit and reach test. The test was performed with the participant in a sitting position on a flat floor with their legs fully extended. The participant’s feet were placed against the base of the sit and reach box with their toes pointing up. They were asked to place one hand over the top of the other hand and bend slowly forward as far as possible, to slide both hands on the measuring board while keeping both knees straight, and to hold the position for at least 2 s. The distance reached was recorded.

Timed up-and-go test (TUG) (s): This test assesses a person’s mobility and balance. Each participant was instructed to complete the course at their usual pace. They had to stand up without any support; walk 3 m, around a cone/mark on the floor, and back to the chair; and then sit down again. The time that the participant needed to complete the test was recorded. A longer time indicated poor balance and mobility performance.

Figure-of-eight walk test (F8W) (s): The F8W, as a measure of walking skill, evaluates movement control and planning during walking. Each participant was instructed to stand midway between two cones placed about 5 ft (1.5 m) apart, start walking at their usual pace in a figure-of-eight walking path around the cones, and stop upon returning to the starting position. Timing began when the participant started to take the first step and ended when both of their feet were back in the starting position.

### 2.5. Exercise Self-Efficacy Scale

Exercise self-efficacy (ESE) was assessed using the scale developed by Marcus et al. [19]. A total of five question items were used to assess the degree of confidence in being able to engage in exercise, such as “I am confident that I can participate in regular exercise even when I feel I don’t have the time”. A five-point Likert scale was used with responses ranging from “I do not think so at all” (1 point) to “I very much think so” (5 points). The scores were totaled for the five scale items as the ESE score, which ranged from 5 to 25 points. A higher score signified higher ESE.

### 2.6. Statistical Analysis

The data of this study were analyzed using the SPSS/PC23.0 (SPSS Inc., Chicago, IL, USA) program. The descriptive statistics on the participant’s general characteristics and study variables were presented as percentage, mean, and standard deviation. For verifying the effectiveness of the application of the augmented-reality-based muscle reduction prevention exercise program, a two-way repeated measures ANOVA was conducted to compare the experimental group and the control group before and after the experiment with a significance level of α = 0.05.

## 3. Results

There was no significant difference observed in the general characteristics between the experimental and control group, confirming homogeneity between the two groups (Table 1).

Looking at the changes in the muscle parameters after the 12-week augmented-reality-based muscle reduction prevention exercise program (Table 3), it was found that the muscle parameters, ASM (kg) (F = 11.222, *p* < 0.002) and SMI (kg/m^2^) (F = 11.222, *p* < 0.002) increased more in the experimental group compared to the control group (F = 11.222, *p* < 0.002), and gait speed was significantly increased (F = 7.221, *p* < 0.005).

Physical performance can be confirmed by the Senior Fitness Test (SFT) (Table 4). After the 12-week program, significant differences were observed in the chair stand test (F = 5.110, *p* < 0.033), the 2-min step test (2MST) (F = 6.621, *p* < 0.020), and the timed up-and-go test (TUG) (F = 5.514, *p* < 0.032).

Finally, exercise self-efficacy was significantly increased in the experimental group (F = 20.464, *p* < 0.001) compared to the control group (Table 5).

## 4. Discussion

In this study, we provided augmented-reality-based muscle reduction prevention exercise programs to elderly women who had used an elderly welfare center, conducted pre- and post-experiment verification between the experimental group and the control group to verify the effectiveness of the program, and then examined the changes in exercise self-efficacy between the two groups to determine the applicability of the augmented-reality-based exercise program.

An elderly welfare center in Korea is one of the leisure facilities for the elderly aged 60 and older and provides various information and services for cultural activities, hobbies, health promotion, and social participation activities for the elderly. As part of their recovery support function, elderly welfare centers run programs to encourage the elderly to participate in exercise to promote their health. As a pilot test that can explore the possibilities of an augmented reality program in line with the fourth industrial era, we used an augmented-reality-based muscle reduction prevention program in this study.

Several methods are used to diagnose sarcopenia. However, the appendicular skeletal muscle mass (ASM) test suggested by Baumgartner [20] in 1998 uses a formula that divides ASM by the square of the height (m^2^) and is the most commonly used test in the Asian Working Group for Sarcopenia (AWGS) [21]. In recent years, however, the European Working Group on Sarcopenia in Older People (EWGSOP) has also included muscle strength (grip strength) and physical performance (walking ability) in the diagnosis of sarcopenia, and these are now commonly used [3].

After the 12-week augmented-reality-based reduction prevention exercise program (Table 4), ASM and SMI muscle parameters were significantly increased in the experimental group compared to the control group. In a study on diabetic patients using 12 weeks of aerobic and muscular strength exercises, it was reported that SMI was significantly increased [22], and in a combined exercise program for the elderly with sarcopenia, it was reported that both ASM and SMI were significantly increased [23]. An increase in SMI is considered to be a response to the increase in ASM with combined exercise, and the increase of SMI in prior studies is consistent with the results of this study.

With aging, cardiopulmonary endurance, joint flexibility, muscle strength, and bone mass decrease, which reduces walking speed, cadence, and heel and arm swing, resulting in unstable gait patterns [24]. It has been confirmed that gait speed and grip strength are reduced in groups with reduced muscle function [25]. However, in a study that applied the “Otago Exercise Program”, which is known as a program suitable for the elderly consisting of muscle strengthening and balancing and walking based on augmented reality, it was reported that there was a significant increase in balance and gait speed (from 0.99 to 0.79 m/s) [26]. In this study, as a result of measuring the 6-m gait speed (m/s), a significant effect was seen in the experimental group, which implies that the gait speed might have been influenced by the ability and coordination of lower extremity muscles enhanced by resistance exercise in patients with sarcopenia [27], supporting the results of this study.

In addition, previous studies have found that the application of an exercise program to prevent muscle reduction and obesity significantly increased the grip strength of the elderly [28]. However, in the current study, using the augmented-reality-based muscle reduction prevention exercise program, a significant increase in the ASM, SMI, and gait speed was found but not in the hand grip strength in the muscle loss index.

One sarcopenia risk factor is restrained physical activity [25]. The physical activity ability of the elderly is shown in their physical strength; the higher the physical strength indicator, the more likely the elderly person is to live a more normal life [29]. On the contrary, skeletal muscle mass reduction due to aging causes a decline in physical activity and daily living ability in the elderly [30]. Therefore, it is very important to prove the effectiveness of physical activity in preventing muscle reduction in the elderly [31].

The Senior Fitness Test (SFT), used as an indicator of physical strength in the elderly, tests lower and upper extremity muscle strength, aerobic endurance, flexibility, agility, and balance and does not require special equipment, so it is widely used to measure the physical fitness of the elderly.

As a result of conducting the SFT for physical activity after the 12-week augmented-reality-based muscle reduction prevention program, significant differences were observed in the chair stand test and the 2-min step test, which is consistent with the results of a study reporting a significant increase in the 2-min step test, chair stand test, and chair sit and reach test after 12 weeks of home-based exercise to prevent muscle reduction [32]. In 6 months of resistance exercise by the elderly aged 82.6 on average, a significant increase was seen in the chair stand test [10], confirming the effectiveness of continued resistance exercise in improving muscle strength and endurance even in the elderly. In addition, the fact that there was a significant difference in the chair stand test and the chair sit and reach test between the active elderly and the elderly with a long sedentary time indicated the importance of maintaining physical strength and participating in exercise on a regular basis to prevent muscle reduction in the elderly [31,33].

Exercise self-efficacy is an important factor contributing to participation in physical activity [34]. In addition, with its advantage of providing experiential and active motivation to users [14,16,17], the augmented reality environment provides a better sense of reality by blending a real environment and virtual objects, thus promoting and inducing rehabilitation in an exercise rehabilitation environment [16]. In fact, it has been reported that augmented-reality-based games have been helpful in the recovery of motor function in joint motion of stroke patients [35], cognition and quality of life in Parkinson’s patients [36], and obesity management through exercise [37].

In addition, the participants can monitor their exercise, promoting rehabilitation in the exercise rehabilitation environment and encouraging exercise [17], which is a meaningful result to the elderly. In particular, considering that the distance between place of residence and place of exercise may restrict the participation of the elderly with sarcopenia who may have experienced a fall from participating in exercise and that exercise participation depends on the season or the environment [38], there is a need to further study the augmented-reality-based exercise program, which is not influenced by the above restrictions, as a meaningful health management program for the elderly.

## 5. Conclusions

Regarding the limitations in this study, in the augmented-reality-based exercise program, the devices were not operated and controlled in all processes by the participants. The exercise specialist set the exercise and executed the program, and the participants exercised by watching the pre-set program through the monitor, checking their posture and accuracy. Therefore, they could see the effect of the exercise more accurately, and there was high exercise self-efficacy through the augmented-reality-based exercise program in this study.

However, the elderly, in reality, have difficulty in adopting information and communications technology [39]. The elderly perceive their age as a factor that hinders them from adopting technology, thus they need operating skills and support. Therefore, with regard to the exercise service through augmented reality devices, in addition to developing the devices, there is a need to train exercise specialists to use and control these devices to support the elderly with exercise.

## Figures and Tables

**Figure 1 ijerph-17-03260-f001:**
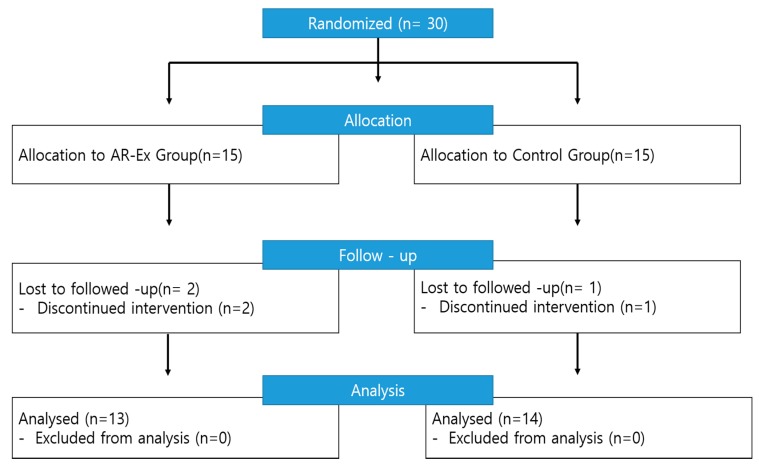
Participant allocation (consolidated standards for reporting of trials flow diagram).

**Figure 2 ijerph-17-03260-f002:**
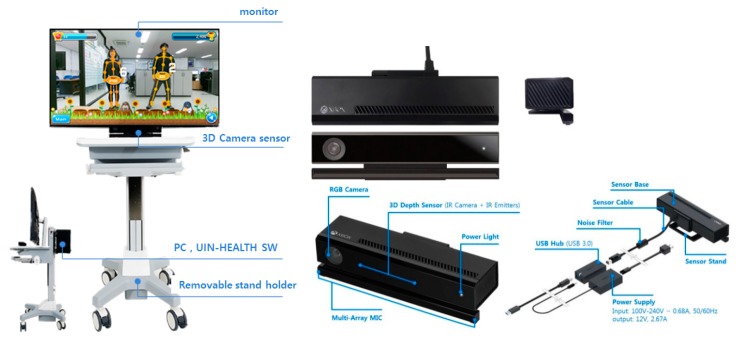
Augmented-reality-based exercise rehabilitation system, UIN-HEALTH^TM^.214.

**Figure 3 ijerph-17-03260-f003:**
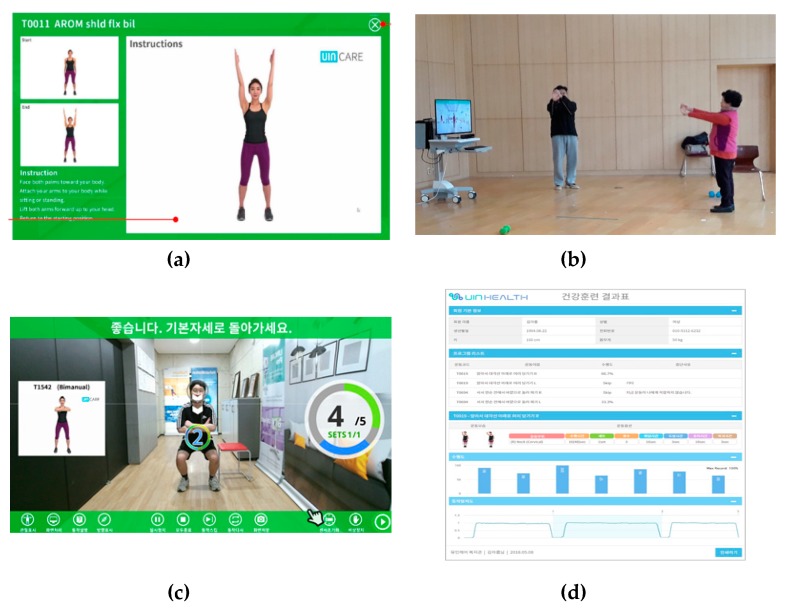
Augmented-reality-based exercise program in UIN-HEALTH. (**a**) Configuration of the contents of the muscle reduction prevention exercise by the exercise specialist; (**b**) content execution and safety management by exercise specialist during exercise; (**c**) exercise situation monitoring on monitors; and (**d**) results of exercise and feedback output.

**Table 1 ijerph-17-03260-t001:** Homogeneity of general characteristics between groups.

Characteristics	Categories	N = 27	AR-EX. (n = 13)	Con. (n = 14)	t or x^2^	*p*
		N (%)	N (%)	N (%)		
Age (year)	65–69	4 (14.8)	2 (15.4)	2 (14.3)	0.038	0.845
	70–74	15 (55.6)	7 (53.8)	8 (57.1)		
	75–79	6 (22.2)	3 (23.1)	3 (21.4)		
	80<	2 (7.4)	1 (7.7)	1 (7.2)		
	M ± SD	72.74 ± 3.64	72.77 ± 3.79	72.71 ± 3.64		
Living together	Yes	7 (25.9)	2 (15.4)	5 (35.7)	1.191	0.020 *
	No	20 (74.1)	11 (84.6)	9 (64.3)		
Education level	None	1 (3.7)	1 (7.7)	0 (0)	−1.262	0.536
	Elementary school	5 (18.6)	3 (23.1)	2 (14.3)		
	Middle school	8 (29.6)	4 (30.8)	4 (28.6)		
	≥High school	13 (48.1)	5 (38.4)	8 (57.1)		
Religion	No	8 (29.6)	4 (30.8)	4 (28.6)	0.123	0.812
	Yes	19 (70.4)	9 (69.2)	10 (71.4)		
Current job	No	16 (59.3)	7 (53.8)	9 (64.3)	−0.534	0.363
	Yes	11 (40.7)	6 (46.2)	5 (35.7)		
Current disease	No	3 (11.1)	1 (7.6)	2 (14.3)	−0.422	0.552
	1	10 (37.0)	6 (46.2)	4 (21.4)		
	≥2	14 (51.9)	6 (46.2)	8 (64.3)		
Body Mass Index (kg/m^2^)	∼18< to <23	3 (11.1)	2 (15.4)	1 (7.1)	−0.120	0.522
	23–25	6 (22.2)	2 (15.4)	4 (28.6)		
	≥25	18 (66.7)	9 (69.2)	9 (64.3)		
	M ± SD	27.78 ± 3.50	27.40 ± 3.46	28.13 ± 3.64		

AR-EX.—augmented-reality-based exercise group; Con.—control group. All data represent mean ± standard deviation. * *p* < 0.05 were analyzed by *t*-test.

**Table 2 ijerph-17-03260-t002:** Program design for augmented-reality-based exercise group.

Type		Program Types	Intensity (RPE)/Time
Warm-up		Stretching	7–9/5 min
Workout	Upper body resistance exercise (5 min)	Shoulder Abduction 180° Shoulder Flexion 180° Shoulder External Rotation A 90° Trunk Lateral Bending 35° Trunk Flexion 80°	9–11/5 min
	Lower body resistance exercise (5 min)	Hip Abduction 45° Hip Flexion 90°, Knee Flexion 90° Sitting Knee Extension 0°	9–11/5 min
	Aerobic exercise (5 min)	Walking in place, forward, sideways, backward Heel raises Knee lifts Leg curls Front lunges	11–13/5 min
	Flexibility exercise (5 min)	Neck Trunk Whole Body	9–11/5 min
Cool-down		Upper body stretching Lower body stretching	7–9/5 min

**Table 3 ijerph-17-03260-t003:** Effects of augmented-reality-based exercise on muscle parameters for older adults.

Categories	Group	Pre-	Post-	ANOVA (*p*)		
		M ± SD	M ± SD	G	T	G × T
ASM (kg)	Exp.	15.32 ± 1.81	15.76 ± 1.67	0.847	0.061	0.003 *
	Con.	15.72 ± 1.62	15.06 ± 1.42			
SMI (kg/m^2^)	Exp.	6.49 ± 0.67	6.69 ± 0.63	0.659	0.030	0.003 *
	Con.	6.71 ± 0.57	6.67 ± 0.53			
Gait speed (m/s)	Exp.	6.98 ± 0.97	6.76 ± 0.89	0.233	0.001	0.013 *
	Con.	7.27 ± 0.73	7.23 ± 0.75			
Hand grip strength (kg)	Exp.	22.55 ± 6.30	22.91 ± 6.16	0.906	0.205	0.109
	Con.	22.41 ± 8.24	22.37 ± 8.37			

All data represent mean ± standard deviation. Exp.—experimental group; Con.—control group; Pre—pretest; Post—posttest; M—mean; SD—standard deviation; G—groups; T—time; G × T— groups × time. * *p* < 0.05 were analyzed by two-way repeated measures ANOVA. ASM—appendicular skeletal muscle mass; SMI—skeletal muscle index.

**Table 4 ijerph-17-03260-t004:** Effects of augmented-reality-based exercise on muscle parameters, physical performance, and exercise self-efficacy for older adults.

Categories	Group	Pre-	Post-	ANOVA (*p*)		
		M ± SD	M ± SD	G	T	G × T
Chair stand test (number in 30 s)	Exp.	20.92 ± 6.59	21.72 ± 5.48	0.795	0.221	0.033 *
	Con.	20.85 ± 5.77	20.62 ± 5.41			
2-min step test (2MST)	Exp.	103.46 ± 7.78	106.00 ± 8.13	0.366	0.339	0.020 *
	Con.	102.42 ± 8.75	101.28 ± 8.43			
Sit and reach test (cm)	Exp.	7.84 ± 15.21	7.22 ± 14.34	0.546	0.492	0.052
	Con.	10.67 ± 13.10	10.96 ± 13.24			
Timed up-and-go test (TUG)	Exp.	6.60 ± 1.39	6.96 ± 1.52	0.730	0.886	0.032 *
	Con.	6.75 ± 1.47	6.43 ± 1.47			
Figure-of-eight walk test (F8W) (s)	Exp.	25.19 ± 5.20	25.53 ± 5.25	0.432	0.319	0.081
	Con.	24.66 ± 3.57	23.45 ± 3.24			

All data represent mean ± standard deviation. * *p* < 0.05 were analyzed by two-way ANOVA.

**Table 5 ijerph-17-03260-t005:** Effects of augmented-reality-based exercise on exercise self-efficacy for older adults.

Categories	Group	Pre-	Post-	ANOVA (*p*)		
		M ± SD	M±SD	G	T	G × T
Exercise self-efficacy	Exp.	14.85 ± 2.67	16.38 ± 3.01	0.766	0.049	0.001 *
	Con.	14.93 ± 2.84	14.36 ± 2.53			

All data represent mean ± standard deviation. * *p* < 0.05 were analyzed by two-way repeated measures ANOVA.

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
