# Peer review of "Effects of Augmented-Reality-Based Exercise on Muscle Parameters, Physical Performance, and Exercise Self-Efficacy for Older Adults"

_ijerph, 2020, doi:10.3390/ijerph17093260_

Round 1
Reviewer 1 Report
As we already know, it is very difficult to conduct research on applying an exercise program which is based on Augmented Reality (AR) technology to elderly people. In that, potentially, this paper can make a useful contribution to a growing body of knowledge on the elderly and AR’s role. Yet, at this stage, a minor revision needs to be made before it is considered as a publishable article in the journal.
Overall, this paper needs to include 1) a revised method section that (a) provides what kind of AR devices have used and the validity of assessment tool and (b) explains how those technologies have been worked, 2) more informed academic discussion on the findings, and 3) some minor typo.
1-a) In the article, I can find an augmented reality-based exercise program which prevents muscle reduction, but do not know what kind of AR devices have been used for the experimental group. Author(s) should explain it because each AR device has a different quality in implementation and in turn, it can affect research results. For example, please give a specific brand and name of device used for research (ex. PlayStation 2 (Sony, Japan) or i-visor FX601 (Daeyang E & C Co., Korea) and, more importantly, provide the validity of assessment tool which affects results.
1-b) At the same time, the author should add whether AR-based exercise program was performed by themselves (participants) or by instructors. It can also affect research results.
2) As author(s) know, a variety of exercise programs have positively affected muscle, physical performance and exercise self-efficacy for the elderly. Here, then the author(s) needs to develop more discussion or argument that highlights how effects of the AR-based exercise program differ from those of previous exercise programs.
3) Other some typo
P5. please explain the meaning of red line in fig. 2
p7. line 7 check SMI (kg). /m2)
p8. line 7 ASM by the square of the height (m2)
Author Response
Thank you for your insightful and useful comments.
We specified the study methods and distinction of AR-based exercise programs in the article.
Point 1:
In the article, I can find an augmented reality-based exercise program which prevents muscle reduction, but do not know what kind of AR devices have been used for the experimental group. Author(s) should explain it because each AR device has a different quality in implementation and in turn, it can affect research results. For example, please give a specific brand and name of device used for research (ex. PlayStation 2 (Sony, Japan) or i-visor FX601 (Daeyang E & C Co., Korea) and, more importantly, provide the validity of assessment tool which affects results.
Response 1:
Thank you for your comment.
Regarding the implementation characteristics AR devices and the devices used in this study, we inserted the details in 2.2. Augmented Reality-based Muscle Reduction Prevention Exercise Program (Line 122-167).
Point 2:
At the same time, the author should add whether AR-based exercise program was performed by themselves (participants) or by instructors. It can also affect research results.
Response 2:
The device was operated by the exercise specialist. We inserted the content in the main text.
Line 152-161.
“The UNI-HEALTH is operated by the exercise specialist providing control of the participant's training modules (content) and the level of difficulty. The experimental group individually participated in exercise at the time appointed with the exercise specialist, and the exercise specialist selected the appropriate exercise for the participant and output safety during exercise and exercise result feedback, serving as an advisor. (Figure 3.).”
Point 3:
As author(s) know, a variety of exercise programs have positively affected muscle, physical performance and exercise self-efficacy for the elderly. Here, then the author(s) needs to develop more discussion or argument that highlights how effects of the AR-based exercise program differ from those of previous exercise programs.
Response 3:
Thank you for your comment
This study was intended to investigate the sustainable exercise of sustainability of the elderly who have little interest in exercise participation or have difficulties in physical activities by motivating them with AR-based exercise program in line with the development of ICT.
As a result of this study, it was found the AR-based exercise program showed a positive effect on the physical fitness and muscle loss index similar to the result of applying general exercise, and significant results were observed in the exercise-efficacy of the elderly. This has been described in more detail in the introduction and the discussions.

Reviewer 2 Report
The subject is very interesting, but I have major concerns regarding English language. There are a lot of English language expression mistakes that should be revised (eg lines 34-36, 41, etc).There are a lot of phrases very difficult to follow because of the bad English language. The manuscript should be revised a..........
- I do not understand what the authors wanted to say in lines 42-44 ....and potential skeletal ....
- line 48, for eg - In my opinion, ”muscle loss” would be more appropiate that ”muscle reduction”
- the ”lack of physical activity” is not a the only cause of developing sarcopenia and this aspect should be clearly specified (line 49).
- line 50 aerobic exercise are.....
- line 49-50 - please rephrase
- Authors should use exercises instead of exercise (aerobic exercises, resistance exercises, etc)
- line 59 - ”....shows them....”. Who is them??
- lines 65-68. Sorry, but I can not follow the phrase. Please rephrase.
- line 76 - please rephrase (what do the author mean by ”women who were using elderly welfare centers”?
- line 83 - What is the purpose of ”Table 1” here?
- line 86 - the same for Fig 2
- Had the author calculated the sample size?
- Have the subjects signed informed consent to participate in the study?
- Was the study approved by the Ethical Committee? The approval number should be mentioned.
- What cut-offs were used in this manuscript for the ASM in order to identify low muscle mass?
- lines 114-116 - the tests used should be described, how are they scored and the interpretation
- Had the author tested if the parameters were normally or non-normally distributed?
- line 138-141 - please rephrase to make the phrase easier to follow
- Did the authors compared the pre and the post results for the Ar-Ex? or just Ar-ex with the control? Lines 138-141
- Table 4 - why was the Mann-Whitney U test used? The p-value was for the Pre- and Post-test comparison? or Ar-Ex compared to controls? The authors should mention that aspect in Table 4
- The authors did not mentioned the EWGSOP 2 diagnostic criteria in lines 160-164.
Author Response
Thank you for your insightful and useful comments.
We specified the research method, corrected the errors in the statistical method, and revised the results and discussions accordingly to strengthen the distinction of the study.
Point 1:
I do not understand what the authors wanted to say in lines 42-44 ....and potential skeletal ....
Response 1:
We corrected as follows:
“and skeletal muscle loss in conjunction with reduction of muscle mass and muscular strength is known to accompanied by a decrease in gait speed as well as a decrease in muscle mass and strength [5].” (Line 44-46).
Point 2:
line 48, for eg - In my opinion, ”muscle loss” would be more appropiate that ”muscle reduction”
the ”lack of physical activity” is not a the only cause of developing sarcopenia and this aspect should be clearly specified (line 49).
line 50 aerobic exercise are.....
line 49-50 - please rephrase
Authors should use exercises instead of exercise (aerobic exercises, resistance exercises, etc)
Response 2:
By reflecting your comments, we inserted references and amended the content as follows.
“and skeletal muscle loss in conjunction with reduction of muscle mass and muscular strength is known to accompanied by a decrease in gait speed as well as a decrease in muscle mass and strength [5].” (Line 45).
“ It not only increases the prevalence of obesity and chronic disease, but also leads to a vicious cycle: physical disorders, poor quality of life and even death [3].”(line 47)
“As sarcopenia is caused by a lack of physical activity with advancing age[6], continued exercise is critical [7]. Aerobic exercise is recommended as a method of managing physical fitness and preventing sarcopenia and aging in the elderly, and resistance exercise is known to be effective not only in maintaining and strengthening muscle functions, but also preventing falls and the impairment of daily living in the elderly [8,9,10].” (Line 50-54)
Point 3:
line 59 - ”....shows them....”. Who is them??
lines 65-68. Sorry, but I can not follow the phrase. Please rephrase.
Response 3:
We corrected the English expression in Line 59 .
“Augmented reality (AR) is a technology that combines the real world and 3D virtual object digital content with additional information into a single image in real time.” (Line 59-60)
We corrected the English expression in Line 65-68.
“In particular, the development and application of the information and communications technology (ICT) in modern society allows it to manage diseases and promote health in an effective way [13], and the AR system is used as a training tool to encourage users to continue rehabilitation through fun and motivation [14].” (Line 63-66)
Point 4:
line 76 - please rephrase (what do the author mean by ”women who were using elderly welfare centers”?
line 83 - What is the purpose of ”Table 1” here?
line 86 - the same for Fig 2
Response 4:
- We corrected Materials and Methods on the whole. (Line 89-237)
We specified the contents of 2.1. Study Design and Participants and changed the positions of figures and tables for easy understanding. (Line 90-120)
2.2. In the Augmented Reality-based Muscle Reduction Prevention Exercise Program, we inserted the details of the operation of the Augmented Reality-based Exercise Program. (Line 122-168)
Point 5:
Had the author calculated the sample size?
Response 5:
The sample size was calculated using G-Power 3.1 program.
We inserted specific content (Line 101-104).
“The sample size was calculated using G-Power 3.1 program. α level, test power, and effect size were set to 0.80 and 0.5, respectively, so the minimum number of participants in this study was 15. However, in this study, 30 participants were recruited considering the dropout of some participants.”
Point 6:
Have the subjects signed informed consent to participate in the study?
Response 6:
We fully explained the participants about the purpose of this study and obtained consents from participants prior to starting this study.
“Except for those who have health problems (chest pain, dizziness, doctor's recommendation on restricted exercise) and those who were not able to understand the purpose of the study due to physical disorders, mental illness and disability in cognition and communication, those who understood the purpose of this study and signed the consent form finally participated in this study.”(Line 97-101)
Point 7:
Was the study approved by the Ethical Committee? The approval number should be mentioned.
Response 7:
We did not proceed with the Ethics Committee approval process for this study, because this study is a pilot test that is part of a research that is being conducted in stages. As a stage to see the reactions of the elderly to the augmented reality-based exercise programs and its effect, this study was conducted with a small number of participants.
After the verification of this stage, we corrected the augmented reality program, simplified the operating section, and developed guidelines for exercise prescriptions. And we proceeded this study stage according to the procedure of the Ethics Committee. Therefore, please understand that there is no approval from the Ethics Committee for this study.
Point 8:
What cut-offs were used in this manuscript for the ASM in order to identify low muscle mass?
Response 8:
We suggested SMI cut-offs in the inclusion criteria of participants.
“Elderly women aged 65 or older who had never participated in physical exercise programs until then were eligible for this study, and among them only those who were above the cut-off value 5.67kg / m2, which is the criterion of the SMI used by the European Working Group on Sarcopenia in Older People (EWGSOP) were included in this study[3].”(Line 94-97)
Point 9:
lines 114-116 - the tests used should be described, how are they scored and the interpretation
Response 9:
We specifically suggested SFT measurement method. (Line 187-222)
Point 10:
Had the author tested if the parameters were normally or non-normally distributed?
Response 10:
The results of this study show the value of the normal distribution. Therefore, we changed the statistical method of this study to two-way ANOVA, a pre- and post-comparison analysis between the two groups.
Therefore, we amended the contents and results in 2.7. Statistical analysis, 3. Results and Tables 3, 4, and 5.(Line 231-250)
Point 11
line 138-141 - please rephrase to make the phrase easier to follow
Did the authors compared the pre and the post results for the Ar-Ex? or just Ar-ex with the control? Lines 138-141
Table 4 - why was the Mann-Whitney U test used? The p-value was for the Pre- and Post-test comparison? or Ar-Ex compared to controls? The authors should mention that aspect in Table 4
Response 11
We suggested the results of the two-way ANOVA according to the revision of the statistical method.
Point 12
The authors did not mentioned the EWGSOP 2 diagnostic criteria in lines 160-164.
Response 12
In the inclusion criteria of participants, we suggested SMI cut-offs applied with EWGSOP 2, and explained. (Line 94-97)
Reviewer 3 Report
Sangwan and colleagues report on the efficacy of an Augmented Reality (AR) exercise intervention in a sample (n=27) of older women who participate in welfare centers in Incheon, Korea. They found that participating in this intervention was associated with increases on several physical functioning measures, including measures of muscle function and physical performance, compared to a control group. They also found a significant treatment effect for exercise self-efficacy. The authors argue that these significant findings provide evidence for the efficacy of the AR intervention in improving physical health of older adults.
I appreciated the interesting intervention and the inclusion of both objective physical measures and perceived self-reported outcome measures. However, I feel that the authors did not provide sufficient details about several aspects of the study design that make it difficult to understand how the intervention was conducted. Furthermore, there are concerns I had about the statistical approach that was taken to test for significant effects, and I believe the findings were overstated given several unacknowledged study limitations (small sample size, generalizability, no-contact control group).
I included both broad and specific comments below separated by section.
Abstract
- Specific
- Use of acronyms (e.g., ASM, SMI) is confusing for those unfamiliar with the measures, please use full names
- Lines 24-25: “significant change was observed in the chair stand test (z=-2.070, p <0.033) and 24 2-min step test (2MST) (z=-2.331, p <0.020)”
- Please specify whether performance improved or declined. It’s currently unclear from the stated z-scores
Introduction
- Broad
- The authors should provide more of a review of how AR interventions have been used for exercise interventions for older adults. Have AR interventions been shown to be feasible and accessible to older adults? Have all of them been effective or just some? How does the current intervention fit in with what has already been done?
- Specific
- Some points require additional clarity. Line 46: please elaborate on the “vicious cycle” between saropenia and disease.
Methods
- Specific
- Line 83: I believe “Table 1” should be “Figure 1” here
- Lines 127-128: If the participants were randomly assigned, the authors do not need to verify the baseline homogeneity of characteristics, because any differences would be due to chance.
- More details or references on the inclusion criteria and random assignment are needed.
- Line 77: What prior exercise programs were considered?
- Line 78: What was the physical strength test used and what was the criterion for passing?
- Line 79: How were participants randomly assigned (e.g., by coin-flip)?
- Broad
- How and when were covariates measured (e.g., age, religion, current diseases)? For religion, current job, and current diseases, please also include what the response options were (i.e., which religions, jobs, and diseases were included)
- More detail on the AR intervention is needed.
- What aspects of the intervention make it an “Augmented Reality” intervention? From Figure 2, it looks like the participant is watching a television, but it is unclear how they are interacting with it. If the intervention is purely spatial AR (Carmigniani et al., 2010, Multimedia Tools and Applications), the authors should specify how it integrates virtual and real environments (e.g., does it track participant movement?).
- Who led the intervention sessions? To what degree were participants allowed to communicate with the session administrators? Did they provide any feedback during the exercises?
- From Figure 2 it appears that an instructor is present guiding the participant through the exercises, in what ways does the digital platform add to the intervention above and beyond the presence of an instructor?
- Perhaps additional screenshots with a better view of the TV screen would help with this
- What degree of adherence to the intervention was needed before an individual was lost to follow-up? Did all individuals who remained in the intervention complete every session?
- More detail on the control group is needed.
- What did the control participants do? Was it a purely passive/no-contact control group?
- If it was a passive control group, and they had no sustained interactions with the researchers as was done in the intervention group, could the intervention effects be in part due to increased physical activity from having to leave the house to attend the exercise sessions 5 days/week rather than the intervention itself?
Results
- Specific
- Line 134: The authors mentioned that there were no significant baseline differences, but “living together” appears to be significant in Table 1 (p=.02).
- This is minor, but it may help to switch the signs on the reported results so that “increases” correspond with positive test statistics.
- Broad
- Please report standard errors, 95% confidence intervals, or some estimate of uncertainty for the test statistics, rather than just the p-values.
- The authors stated that they used independent samples t-tests and Mann-Whitney U tests to test for intervention effects, but then list z-statistics in their results instead of t-statistics and U statistics. Please correct this or clarify the tests used.
- More importantly, if I understand them correctly, the current analyses do not account for baseline differences in the outcomes, but just examine the changes over time, leading to potentially biased treatment effects (Twisk et al. 2018, Contemporary Clinical Trials Communications). Please either adjust for baseline outcomes or use an alternative approach that accounts for this (e.g., ANCOVA, repeated measures analyses).
Discussion
- Specific
- The first paragraph of the discussion has information about the physical measures included that may be better to include in the introduction or methods section
- Test statistics and p-values should not be included in the discussion.
- Lines 184-185: Please clarify whether or not the experimental group actually did increase in grip strength.
- Broad
- The authors introduce and reference the Otago exercise platform in the discussion, but it would be best to include this in the introduction and methods, with a more detailed description on how this intervention works in these sections. As a reader, I did not realize this was an established exercise platform until it was mentioned in the discussion.
- While the authors report multiple significant associations between groups, are the increases within the treatment group significant and clinically meaningful (e.g., the .44 increase in ASM)?
- The authors overstate their findings without including mention of study limitations, some of which (included below) greatly limit the conclusions that can be drawn from their current results.
- Sample size: The small sample size limits the statistical power to precisely estimate effect sizes. This is further complicated by the lack of confidence intervals or measures of uncertainty for the stated results.
- Generalizability: This study was limited to older women who participated in welfare centers. Furthermore, these women may have been more motivated to participate in an intervention involving technology than women in the general population, which may have correlates with personality or other individual differences (Vroman et al., 2015, Computers in Human Behavior).
- Passive control group: No way to determine whether it was the exercise intervention itself or some other aspect related to participating in the intervention (e.g., travel, social engagement, etc.) that is driving the effect.
- The authors should expand on how the AR environment provides motivation to users (Line 210), given this appears to be a key factor for how AR contributes to exercise interventions.
- The authors report an interesting, significant finding with exercise self-efficacy. What are the hypothesized mechanisms for this relationship, as informed by the literature? I think these are important to mention as they could be explored further in future studies, and they provide additional rationale for why AR is beneficial to exercise interventions.
Tables and Figures
- Table 1
- Please clarify what “living together” means. Could the authors perhaps mean “living with others” (vs. living alone)?
- Table 2
- Please specify what “RPE” is.
- Table 4
- Please specify that the test statistics reported are for the pre-post differences in measures between the intervention and control groups, not differences in single scores.
- Please provide estimates and standard deviations for pre-post differences for both AR-EX and Control groups so that the test statistics can be confirmed.
Author Response
Thank you for your insightful and useful comments.
We specified the research method, corrected the errors in the statistical method, and revised the results and discussions accordingly to strengthen the distinction of the study.
Point 1: Use of acronyms (e.g., ASM, SMI) is confusing for those unfamiliar with the measures, please use full names.
Lines 24-25: “significant change was observed in the chair stand test (z=-2.070, p <0.033) and 24 2-min step test (2MST) (z=-2.331, p <0.020)”
Please specify whether performance improved or declined. It’s currently unclear from the stated z-scores
Response 1:
We inserted the full name of ASM and SMI.
We corrected the statistical methods and results accordingly. (Line 21)
“Appendicular skeletal muscle mass (ASM) (F = 11.222, p <0.002) and Skeletal muscle index (SMI)”
Point 2: Introduction
The authors should provide more of a review of how AR interventions have been used for exercise interventions for older adults. Have AR interventions been shown to be feasible and accessible to older adults? Have all of them been effective or just some? How does the current intervention fit in with what has already been done?
Response 2:
By reflecting the contents, we inserted previous studies on how AR intervention affected the elderly.(Line 63-78)
“In particular, the development and application of the information and communications technology (ICT) in modern society allows it to manage diseases and promote health in an effective way [13], and the AR system is used as a training tool to encourage users to continue rehabilitation through fun and motivation [14]. AR technology, in particular, is widely used in the rehabilitation of those with diseases such as stroke because three factors of rehabilitation (repetition, rapid feedback, and motivation) conveyed through augmented reality are found to be very effective [15].
Through the augmented reality, the participants became interested and motivated to exercise, and access to and control of the exercise became easy. As a result of the exercise, they could improve muscle strength, range of joint motion, speed of exercise, and ability to control exercise [14].The feedback function of virtual reality is very helpful in retraining the movement of uses by displaying information about errors in the user's movement and tracking of correct movement [16]. In addition, the AR, a field of the virtual reality, is a hybrid VR system that synthesizes virtual objects in a real environment and makes them look like the objects existing in the original environment, and the augmented reality technology that mixes real environment and virtual objects can provide a better sense of reality by allowing users to see the real environment, thus promoting and inducing rehabilitation in a real exercise rehabilitation environment [17].”
Point 3: Some points require additional clarity. Line 46: please elaborate on the “vicious cycle” between saropenia and disease.
Response 3:
We corrected the expression of the contents as follows.(Line 47-49)
“The elderly with sarcopenia are up to 1.5 times more likely to have cancer and stroke. It not only increases the prevalence of obesity and chronic disease, but also leads to a vicious cycle: physical disorders, poor quality of life and even death [3].”
Point 4: Methods
Line 83: I believe “Table 1” should be “Figure 1” here
Lines 127-128: If the participants were randomly assigned, the authors do not need to verify the baseline homogeneity of characteristics, because any differences would be due to chance.
More details or references on the inclusion criteria and random assignment are needed.
Line 77: What prior exercise programs were considered?
Line 78: What was the physical strength test used and what was the criterion for passing?
Line 79: How were participants randomly assigned (e.g., by coin-flip)?
Response 4:
Line 83: We corrected the main text and inserted the content in 2.2. Augmented Reality-based Muscle Reduction Prevention Exercise Program .(Line 122-167)
Lines 127-128: We corrected the content in 2.7. Modified in Statistical analysis. (Line 231-237)
Line 77: We corrected the English expression. (Line 94-95)
“Elderly women aged 65 or older who had never participated in physical exercise programs until then were eligible for this study ”
Line 78: We inserted the content. (Line 97-100)
“Except for those who have health problems (chest pain, dizziness, doctor's recommendation on restricted exercise) and those who were not able to understand the purpose of the study due to physical disorders, mental illness and disability in cognition and communication,”
Line 79: We inserted the content. (Line 104-108)
“Participants in the study were 30 elderly women who had attended the elderly welfare centers located in Incheon City from Jan 7 to Jan 18, 2019 and randomly selected the allocation ratio by SPSS Win23.0 program. They were randomized to the two groups-the experimental group (15 participants) treated with the augmented reality-based exercise and the control group (15 participants) – at the allocation ratio of 50.0%.”
Point 5:
How and when were covariates measured (e.g., age, religion, current diseases)? For religion, current job, and current diseases, please also include what the response options were (i.e., which religions, jobs, and diseases were included)
More detail on the AR intervention is needed.
What aspects of the intervention make it an “Augmented Reality” intervention? From Figure 2, it looks like the participant is watching a television, but it is unclear how they are interacting with it. If the intervention is purely spatial AR (Carmigniani et al., 2010, Multimedia Tools and Applications), the authors should specify how it integrates virtual and real environments (e.g., does it track participant movement?).
Who led the intervention sessions? To what degree were participants allowed to communicate with the session administrators? Did they provide any feedback during the exercises?
From Figure 2 it appears that an instructor is present guiding the participant through the exercises, in what ways does the digital platform add to the intervention above and beyond the presence of an instructor?
Perhaps additional screenshots with a better view of the TV screen would help with this
What degree of adherence to the intervention was needed before an individual was lost to follow-up? Did all individuals who remained in the intervention complete every session?
More detail on the control group is needed.
What did the control participants do? Was it a purely passive/no-contact control group?
If it was a passive control group, and they had no sustained interactions with the researchers as was done in the intervention group, could the intervention effects be in part due to increased physical activity from having to leave the house to attend the exercise sessions 5 days/week rather than the intervention itself?
Response 5:
We inserted detailed method in 2.2. Augmented Reality-based Muscle Reduction Prevention Exercise Program.(Line 122-167)
Point 6: Results
Line 134: The authors mentioned that there were no significant baseline differences, but “living together” appears to be significant in Table 1 (p=.02).
This is minor, but it may help to switch the signs on the reported results so that “increases” correspond with positive test statistics.
Response 6:
We showed significances in Table 1.(Line 118)
Point 7: Results
Please report standard errors, 95% confidence intervals, or some estimate of uncertainty for the test statistics, rather than just the p-values.
The authors stated that they used independent samples t-tests and Mann-Whitney U tests to test for intervention effects, but then list z-statistics in their results instead of t-statistics and U statistics. Please correct this or clarify the tests used.
More importantly, if I understand them correctly, the current analyses do not account for baseline differences in the outcomes, but just examine the changes over time, leading to potentially biased treatment effects (Twisk et al. 2018, Contemporary Clinical Trials Communications). Please either adjust for baseline outcomes or use an alternative approach that accounts for this (e.g., ANCOVA, repeated measures analyses).
Response 7:
I admit that there were errors in the statistics processing. We conducted analysis using two-way ANCOVA for the pre- and post-comparison analysis between the two groups, and corrected results, statistical method and discussions on the whole. and then corrected 2.7. Statistical analysis (Line 231-237) and 3. Results accordingly in the main text. (Line 238-250)
Point 8: Discussion
The first paragraph of the discussion has information about the physical measures included that may be better to include in the introduction or methods section
Test statistics and p-values should not be included in the discussion.
Lines 184-185: Please clarify whether or not the experimental group actually did increase in grip strength.
Response 8:
We corrected the first paragraph. (Line 265-276)
“In this study, we provided augmented reality-based muscle reduction prevention exercise programs to the elderly women using the elderly welfare center; conducted pre-post verification between the experimental group and the control group to verify the effectiveness of the program; and then examined changes in exercise self-efficacy between the two groups to determine the applicability of augmented reality-based exercise program.
The elderly welfare center in Korea is one of the leisure and leisure facilities for the elderly aged 60 and older, and provides various information and services for cultural activities, hobbies, health promotion and social participation activities of the elderly. As part of its function recovery support business, the elderly welfare center is running a program to encourage the elderly to participate in exercise to promote their health. As a pilot test that can explore the possibilities of the augmented reality program in line with the 4th industrial era, we applied the augmented reality-based muscle reduction prevention program in this study.”
We deleted all p-values from the main text.
There was no significant change in grip strength. We corrected the English expression.(Line 302-304)
“However, it was found that there was a significant increase in the remaining ASM, SMI, and Gait speed except for the hand grip strength in the muscle loss index, which is the application of the augmented reality-based muscle reduction prevention exercise program in this study.”
Point 9:
The authors introduce and reference the Otago exercise platform in the discussion, but it would be best to include this in the introduction and methods, with a more detailed description on how this intervention works in these sections. As a reader, I did not realize this was an established exercise platform until it was mentioned in the discussion.
Response 9:
We inserted the description of Otago Exercise. (Line 293-296)
“However, in a study that applied ‘Otago Exercise Program’ which is known as a program suitable for the elderly consisting of muscle strengthening and balancing and walking based on the augmented reality, it was reported that there was a significant increase in balance and gait speed (from 0.99m/s to 0.79m/s) [26].”
Point 10:
While the authors report multiple significant associations between groups, are the increases within the treatment group significant and clinically meaningful (e.g., the .44 increase in ASM)?
The authors overstate their findings without including mention of study limitations, some of which (included below) greatly limit the conclusions that can be drawn from their current results.
Sample size: The small sample size limits the statistical power to precisely estimate effect sizes. This is further complicated by the lack of confidence intervals or measures of uncertainty for the stated results.
Response 10:
We discussed through the correction of statistical method and its results.
Please check the corrected content again.
Point 11:
Generalizability: This study was limited to older women who participated in welfare centers. Furthermore, these women may have been more motivated to participate in an intervention involving technology than women in the general population, which may have correlates with personality or other individual differences (Vroman et al., 2015, Computers in Human Behavior).
Passive control group: No way to determine whether it was the exercise intervention itself or some other aspect related to participating in the intervention (e.g., travel, social engagement, etc.) that is driving the effect.
Response 11:
Regarding the background of this study conducted with the elderly attending the elderly welfare centers, we inserted the content as follows (Line 270-276).
“The elderly welfare center in Korea is one of the leisure and leisure facilities for the elderly aged 60 and older, and provides various information and services for cultural activities, hobbies, health promotion and social participation activities of the elderly. As part of its function recovery support business, the elderly welfare center is running a program to encourage the elderly to participate in exercise to promote their health. As a pilot test that can explore the possibilities of the augmented reality program in line with the 4th industrial era, we applied the augmented reality-based muscle reduction prevention program in this study. “
We also agree with the suggestions in the study of ‘Computers in Human Behavior’ (Vroman et al., 2015). We are conducting other studies (participation and questionnaire) to this effect.
Nevertheless, this study is a pilot test for exploring the possibility of introducing ICT-based exercise programs in response to changes in 4th Industrial Revolution.
This study is focused on exploring the effects of the exercise and the possibility of participation by the elderly when the functional role of exercise is realized as ICT-based augmented reality.
Point 12:
The authors should expand on how the AR environment provides motivation to users (Line 210), given this appears to be a key factor for how AR contributes to exercise interventions.
Response 12:
We inserted the contents in the introduction that the content of Reference 14-16-17 can be applied. (Line 69-78) (Line 325-341)
Point 13:
The authors report an interesting, significant finding with exercise self-efficacy. What are the hypothesized mechanisms for this relationship, as informed by the literature? I think these are important to mention as they could be explored further in future studies, and they provide additional rationale for why AR is beneficial to exercise interventions.
Response 13:
We inserted the mechanism of AR-based exercise efficacy in the discussion. (Line 325-341)
“Given that the exercise self-efficacy is an important factor that contributes to inducing the elderly to participate in physical activity [34], the augmented reality-based exercise program of this study provided a better realism by mixing real environment and virtual objects, and the participants could monitor their exercise, promoting rehabilitation in the exercise rehabilitation environment and encouraging exercise [17], which is meaningful result to the elderly. In particular, considering that the distance between place of residence and exercise place may restrain the elderly with sarcopenia who experienced a fall from participating in exercise and that exercise participation depends on the season or the environment [38], there is a need to study further the augmented reality-based exercise program, which is not influenced by the above restraints, as meaningful health management program for the elderly.“
At the end of the discussion, we suggested the limitations of this study and directions for future studies (Line 342-353).
“Regarding the limitations in this study, in the augmented reality-based exercise program, devices were not operated and controlled in all processes by participants. The exercise specialist set the exercise and executed the program, and participants exercised by watching the pre-set program through the monitor and checked their posture and accuracy. Therefore, they could see the effect of exercise more accurately, and high exercise self-efficacy through the augmented reality-based exercise program in this study.
However, the elderly, in reality, are having difficulties adopting information and communications technology [39]. The elderly perceive their age as a factor that hinders them form adopting technology, thus wanting operating skills and supports. Therefore, with regard to the exercise service through the augmented reality devices, there is a need to introduce and train the exercise specialist who can prescribe by controlling these devices as well as to develop the augmented reality device.“

Round 2
Reviewer 2 Report
All my concerns have been answered. The manuscript was significantly improved.
Author Response
Thank you for your insightful and useful comments.
We correct the entire English language editing by MDPI.
Attach the file of the last modified report.

Reviewer 3 Report
I appreciate the response the authors made to my comments, especially the expanded methods section with more details about the AR intervention and sample, changes in the statistical methods, and changes to the discussion.
I think most of my comments were addressed adequately and the paper is much improved.
The following, if addressed, should be sufficient for making the paper acceptable:
- Please state/clarify explicitly in the methods that the control group was not asked to do anything. Intervention research is increasing using both passive (i.e., no contact) and active (i.e., alternative intervention) control groups, so clarifying this is important.
- Before publication, I think the English language and style of the paper needs some attention to improve the clarity and flow.
Author Response
Thank you for your insightful and useful comments.
We specified the research method, corrected the errors in the statistical method, and revised the results and discussions accordingly to strengthen the distinction of the study.
Point 1: Please state/clarify explicitly in the methods that the control group was not asked to do anything. Intervention research is increasing using both passive (i.e., no contact) and active (i.e., alternative intervention) control groups, so clarifying this is important.
Response 1:
We inserted detailed method in Line 109-111.
“The control group was noncontact other than the participants being tested pre- and post-experiment, and participants in this group were asked not to do any exercise during the experiment.”
Point 2: Before publication, I think the English language and style of the paper needs some attention to improve the clarity and flow.
Response 2:
Thank you for your insightful and useful comments.
We correct the entire English.
Attach the file of the last modified report.
